# Study on Structural Design and Analysis of Composite Boat Hull Manufactured by Resin Infusion Simulation

**DOI:** 10.3390/ma14205918

**Published:** 2021-10-09

**Authors:** Haseung Lee, Kyungwoo Jung, Hyunbum Park

**Affiliations:** 1School of Mechanical Convergence System Engineering, Kunsan National University, 558 Daehak-ro, Miryong-dong, Gunsan 54150, Jeollabuk-do, Korea; imoger@nate.com; 2S Comtech Inc., 8 Sudang-gil, Chosung-meon, Boseong 59440, Jeollanam-do, Korea; scomtech+yacht@naver.com

**Keywords:** analytical modeling, numerical properties, resin flow

## Abstract

In this paper, structural design and analysis of a composite boat hull was performed. A resin transfer molding manufacturing method was adopted for manufacturing the composite boat hull. The RTM process is an advanced composite manufacturing method that allows a much higher quality product than the hand lay-up process, and less manufacturing cost compared to the autoclave method. Therefore, the RTM manufacturing method was adopted. The mechanical properties of the various aramid fibers and polyester resin were investigated. Based on this, structural design of boat hull was performed using aramid fiber or polyester. After structural design, the optimized resin infusion analysis for RTM manufacturing method was performed. Through the resin infusion analysis, it is confirmed that the designed location of resin injection and outlet is acceptable for manufacturing.

## 1. Introduction

The composite materials are widely used in vehicle design because of their excellent strength and light weight. Although the aramid fiber has less strength than the high strength fiber such as the carbon fiber, it has low manufacturing cost. Accordingly, it can be applied as very advantageous composite when an appropriate resin has been selected. In this work, structural design and analysis of composite boat hull was performed using aramid fiber or polyester.

Among the previous studies, F. M. Santos et al. conducted modal analysis of a fast patrol boat made of composite material. In this paper, the structural dynamic behavior of a fast patrol boat is studied [1].

A. J. Sobey et al. performed optimization of composite boat hulls using first principles and design rules. In this work, a method for rapid optimization of composite grillage structures is proposed [2].

Yongkuk Jeong et al. performed development of the methodology for environmental impact of composite boats manufacturing process. In this paper, environmental impact evaluation system was developed to calculate the environmental impact from the composite boat [3].

S. Castegnaro et al. conducted study on design and manufacturing of bio-composite racing sailboat. In this work, the development of a flax-epoxy and balsa wood racing sailboat was presented, from the materials selection to the manufacturing technique [4].

Richard Pemberton et al. studied investigation on structural composite for marine boat constructions. In this study, the properties of an alternative material for use in marine engineering was investigated [5].

Adrian Caramatescu et al. performed the experimental and numerical evaluation of wave impact stress on a composite boat hull. This paper focused on the finite element analysis for impact [6]. 

N. Baral et al. conducted study on improved impact performance of marine sandwich panels using through-thickness reinforcement experimentally. This paper presents results from a test developed to simulate the water impact (slamming) loading of sandwich boat structures [7].

Sang-Young Kim et al. performed study on mechanical properties test of hand-layup and vacuum infusion processed hybrid composite materials for GFRP marine structures [8].

Many research works of structural design and analysis using a composite boat were performed. However, little research work has been carried out to analysis resin injection and outlet for RTM manufacturing. Additionally, also, previous studies did not consider the research of optimized RTM manufacturing process.

In this study, a composite was applied to conduct the structural design and analysis of small boats. For the manufacturing method, a resin transfer molding method was adopted. Various design variables were analyzed for resin infiltration of the resin transfer molding (RTM) manufacturing method. On the basis of that, the optimized positions of inlet and outlet were determined through the RTM manufacturing method simulation. The final resin infusion simulation result and the manufacturing result were compared to verify the validity of analysis techniques.

## 2. Structural Design and Analysis

This study determined the thickness and lay-up sequence of structures through the hull structure design. For structural design loads, the hull weight and payload were investigated. The buoyancy force due to water pressure was applied as an external load. The buoyant force is the greatest at the bottom of a hull. Figure 1 shows schematic diagram of static water pressure. The pressure increases as the water deepens. The pressure rises by about 1kg/cm^2^ per 10 m. If there is no payload, the more bending load works. In addition, in the case where a moment works, if a ship is tilting sharply to one side, the moment works inward, and to support it, the transverse bulkhead and transverse hatch coaming should be strong [9]. Figure 2 shows load diagram of hull applied to moment.

The specification of the boat to be studied is a small boat, which is 5 m in hull length, 2.3 m in width and 30 knot. Figure 3 shows the configuration of hull structure. Figure 4 shows dimension of boat hull. The structural design was conducted by considering the structural load when the payload works on the bottom of the hull to make the maximum buoyancy and moment. The structural design method with composite laminates was applied to determine the lay-up pattern and thickness of the hull structure. 

The structural design technique with composites was a simple design one, which performed conceptual design. The simple design result was complemented by applying the rule of mixture design technique and then the lamination and thickness were supplemented. In order to simplify the design, the laminate netting rule, which is assuming that only fiber direction layers can provide the stiffness, i.e., no stiffness contribution from off-axis layers, is firstly applied to determine the thickness of the laminate structure. The principal stress design method is used because of the reduction in weight and well-defined load directions.

The final design was performed by laminate constitutive theory. Laminates used in most engineering applications are fabricated by stacking plies in different orientations. A commonly used method of determining stresses and strains for such laminates is based on the classical laminate plate theory. A lamina is thin compared to other dimensions of the entire laminate. Therefore, the lamina can be assumed to be in a state of generalized plane stress. Consequently, all the through-thickness stress component are zero, i.e., *σ*_4_ = *σ*_5_ = *σ*_6_ = 0. In such a case, the constitutive relation for an individual lamina referred to the three axes of symmetry can be written in Voigt notation as
(1)$$\left\{\begin{array}{c} \begin{array}{c} \sigma_{1}\\ \sigma_{2} \end{array}\\ \sigma_{6} \end{array}\right\}=\left[\begin{array}{ccc} Q_{11} & Q_{12} & 0\\ Q_{12} & Q_{22} & 0\\ 0 & 0 & Q_{66} \end{array}\right]\left\{\begin{array}{c} \varepsilon_{1}\\ \varepsilon_{2}\\ \varepsilon_{6} \end{array}\right\}$$
$$Q_{11}=\frac{E_{1}}{1-\nu_{12}\nu_{21}},\;Q_{22}=\frac{E_{2}}{1-\nu_{12}\nu_{21}},$$
$$Q_{12}=\frac{\nu_{12}E_{2}}{1-\nu_{12}\nu_{21}}=\frac{\nu_{21}E_{1}}{1-\nu_{12}\nu_{21}},$$
(2)$$Q_{66}=G_{12}$$

The inverse constitutive relation for the lamina is given by
(3)$$\varepsilon_{ij}=S_{ijkl}\sigma_{kl}=\left\{\begin{array}{c} \varepsilon_{1}\\ \varepsilon_{2}\\ \varepsilon_{6} \end{array}\right\}=\left[\begin{array}{ccc} \frac{1}{E_{1}} & -\frac{\nu_{21}}{E_{2}} & 0\\ -\frac{\nu_{12}}{E_{1}} & \frac{1}{E_{2}} & 0\\ 0 & 0 & \frac{1}{G_{12}} \end{array}\right]\left\{\begin{array}{c} \begin{array}{c} \sigma_{1}\\ \sigma_{2} \end{array}\\ \sigma_{6} \end{array}\right\}$$

The above constitutive relations are written in the lamina coordinate system (i.e., with *x*_1_ along the fiber direction, *x*_2_ normal to the fiber direction, and *x*_3_ along the lamina thickness). The constitutive relation for the lamina in another coordinate system (x-y-z), which, for instance, could be aligned with the coordinate system chosen for the laminate is
(4)$$\left\{\begin{array}{c} \begin{array}{c} \sigma_{xx}\\ \sigma_{yy} \end{array}\\ \sigma_{xy} \end{array}\right\}=\left[\begin{array}{ccc} {\bar{Q}}_{11} & {\bar{Q}}_{12} & {\bar{Q}}_{16}\\ {\bar{Q}}_{12} & {\bar{Q}}_{22} & {\bar{Q}}_{26}\\ {\bar{Q}}_{16} & {\bar{Q}}_{26} & {\bar{Q}}_{66} \end{array}\right]\left\{\begin{array}{c} \varepsilon_{xx}\\ \varepsilon_{yy}\\ 2\varepsilon_{xy} \end{array}\right\}$$
where ${\bar{Q}}_{ij}$ are known as reduced stiffness coefficients. These are related to $Q_{ij}$, defined by Equation (2), by the transformation rules for stresses and strains.

Using the lamina constitutive relations described earlier, the constitutive equation for the kth(*k* = 1, 2, …) layer of the laminate can be written as
(5)$${\left\{\sigma \right\}}^{\left(k\right)}={\left[\bar{Q}\right]}^{\left(k\right)}{\left\{\varepsilon \right\}}^{\left(k\right)}$$

In the above equation, the square bracket represents a 3 × 3 matrix and the curly bracket is for a 3 × 1 vector. The strains in the kth ply are given by
(6)$${\left\{\varepsilon \right\}}^{\left(k\right)}=\left\{\varepsilon^{0}\right\}+z\left\{k\right\}$$

The thermal strains can be added to these strains, such that
(7)$${\left\{\varepsilon \right\}}^{\left(k\right)}=\left\{\varepsilon^{0}\right\}+z\left\{k\right\}-{\left\{\alpha \right\}}_{k}∆T$$

The kth ply stresses on using Equation (5) can now be written as
(8)$${\left\{\sigma \right\}}^{\left(k\right)}={\left[\bar{Q}\right]}^{\left(k\right)}\left(\left\{\varepsilon^{0}\right\}-{\left\{\alpha \right\}}_{k}∆T\right)+z{\left[\bar{Q}\right]}^{\left(k\right)}\left\{k\right\}$$

At the laminate level the force and moment resultants are defined as
(9)$$\left[N\right]=\left\{\begin{array}{c} N_{xx}\\ N_{yy}\\ N_{xy} \end{array}\right\}=\int_{-h/2}^{h/2}\left\{\begin{array}{c} \sigma_{xx}\\ \sigma_{yy}\\ \sigma_{xy} \end{array}\right\}dz$$
$$\left[M\right]=\left\{\begin{array}{c} M_{xx}\\ M_{yy}\\ M_{xy} \end{array}\right\}=\int_{-h/2}^{h/2}\left\{\begin{array}{c} \sigma_{xx}\\ \sigma_{yy}\\ \sigma_{xy} \end{array}\right\}dz$$

In terms of ply stresses that generally vary from ply to ply, we have
(10)$$\left\{\begin{array}{c} \begin{array}{c} N_{xx}\\ N_{yy} \end{array}\\ N_{xy} \end{array}\right\}=\overset{N}{\underset{k=1}{\sum }}\int_{z_{k}}^{z_{k+1}}\left\{\begin{array}{c} \sigma_{xx}\\ \sigma_{yy}\\ \sigma_{xy} \end{array}\right\}dz$$

Which given us
(11)$$\left\{\begin{array}{c} N_{xx}\\ N_{yy}\\ N_{xy} \end{array}\right\}+\left\{\begin{array}{c} N_{xx}^{th}\\ N_{yy}^{th}\\ N_{xy}^{th} \end{array}\right\}=$$
$$\overset{N}{\underset{k=1}{\sum }}\int_{z_{k}}^{z_{k+1}}\left[\begin{array}{ccc} {\bar{Q}}_{11} & {\bar{Q}}_{12} & {\bar{Q}}_{16}\\ {\bar{Q}}_{12} & {\bar{Q}}_{22} & {\bar{Q}}_{26}\\ {\bar{Q}}_{16} & {\bar{Q}}_{26} & {\bar{Q}}_{66} \end{array}\right]\left\{\begin{array}{c} \varepsilon_{xx}^{0}\\ \varepsilon_{yy}^{0}\\ \gamma_{xy}^{0} \end{array}\right\}dz+\overset{N}{\underset{k=1}{\sum }}\int_{z_{k}}^{z_{k+1}}\left[\begin{array}{ccc} {\bar{Q}}_{11} & {\bar{Q}}_{12} & {\bar{Q}}_{16}\\ {\bar{Q}}_{12} & {\bar{Q}}_{22} & {\bar{Q}}_{26}\\ {\bar{Q}}_{16} & {\bar{Q}}_{26} & {\bar{Q}}_{66} \end{array}\right]\left\{\begin{array}{c} k_{xx}\\ k_{yy}\\ k_{xy} \end{array}\right\}zdz$$
where the force resultants due to thermal stresses are given by
(12)$$\left\{\begin{array}{c} N_{xx}^{th}\\ N_{yy}^{th}\\ N_{xy}^{th} \end{array}\right\}=\overset{N}{\underset{k=1}{\sum }}\int_{z_{k}}^{z_{k+1}}\left[\begin{array}{ccc} {\bar{Q}}_{11} & {\bar{Q}}_{12} & {\bar{Q}}_{16}\\ {\bar{Q}}_{12} & {\bar{Q}}_{22} & {\bar{Q}}_{26}\\ {\bar{Q}}_{16} & {\bar{Q}}_{26} & {\bar{Q}}_{66} \end{array}\right]\left\{\begin{array}{c} \alpha_{x}∆T\\ \alpha_{y}∆T\\ 0 \end{array}\right\}dz$$

The relation in Equation (11) can be rewritten in more compact form by using matrices [A] and [B] as follows
(13)$$\left\{\begin{array}{c} N_{xx}\\ N_{yy}\\ N_{xy} \end{array}\right\}+\left\{\begin{array}{c} N_{xx}^{th}\\ N_{yy}^{th}\\ N_{xy}^{th} \end{array}\right\}=\left[\begin{array}{ccc} A_{11} & A_{12} & A_{16}\\ A_{12} & A_{22} & A_{26}\\ A_{16} & A_{26} & A_{66} \end{array}\right]\left\{\begin{array}{c} \varepsilon_{xx}^{0}\\ \varepsilon_{yy}^{0}\\ \gamma_{xy}^{0} \end{array}\right\}+\left[\begin{array}{ccc} B_{11} & B_{12} & B_{16}\\ B_{12} & B_{22} & B_{26}\\ B_{16} & B_{26} & B_{66} \end{array}\right]\left\{\begin{array}{c} k_{xx}\\ k_{yy}\\ k_{xy} \end{array}\right\}$$

Similarly, moment equation is like this.
(14)$$\left\{\begin{array}{c} \begin{array}{c} M_{xx}\\ M_{yy} \end{array}\\ M_{xy} \end{array}\right\}=\overset{N}{\underset{k=1}{\sum }}\int_{z_{k}}^{z_{k+1}}\left\{\begin{array}{c} \sigma_{xx}\\ \sigma_{yy}\\ \sigma_{xy} \end{array}\right\}zdz$$
(15)$$\left\{\begin{array}{c} M_{xx}\\ M_{yy}\\ M_{xy} \end{array}\right\}+\left\{\begin{array}{c} M_{xx}^{th}\\ M_{yy}^{th}\\ M_{xy}^{th} \end{array}\right\}=$$
$$\overset{N}{\underset{k=1}{\sum }}\int_{z_{k}}^{z_{k+1}}\left[\begin{array}{ccc} {\bar{Q}}_{11} & {\bar{Q}}_{12} & {\bar{Q}}_{16}\\ {\bar{Q}}_{12} & {\bar{Q}}_{22} & {\bar{Q}}_{26}\\ {\bar{Q}}_{16} & {\bar{Q}}_{26} & {\bar{Q}}_{66} \end{array}\right]\left\{\begin{array}{c} \varepsilon_{xx}^{0}\\ \varepsilon_{yy}^{0}\\ \gamma_{xy}^{0} \end{array}\right\}zdz+\overset{N}{\underset{k=1}{\sum }}\int_{z_{k}}^{z_{k+1}}\left[\begin{array}{ccc} {\bar{Q}}_{11} & {\bar{Q}}_{12} & {\bar{Q}}_{16}\\ {\bar{Q}}_{12} & {\bar{Q}}_{22} & {\bar{Q}}_{26}\\ {\bar{Q}}_{16} & {\bar{Q}}_{26} & {\bar{Q}}_{66} \end{array}\right]\left\{\begin{array}{c} k_{xx}\\ k_{yy}\\ k_{xy} \end{array}\right\}z^{2}dz$$
where
(16)$$\left\{\begin{array}{c} M_{xx}^{th}\\ M_{yy}^{th}\\ M_{xy}^{th} \end{array}\right\}=\overset{N}{\underset{k=1}{\sum }}\int_{z_{k}}^{z_{k+1}}\left[\begin{array}{ccc} {\bar{Q}}_{11} & {\bar{Q}}_{12} & {\bar{Q}}_{16}\\ {\bar{Q}}_{12} & {\bar{Q}}_{22} & {\bar{Q}}_{26}\\ {\bar{Q}}_{16} & {\bar{Q}}_{26} & {\bar{Q}}_{66} \end{array}\right]\left\{\begin{array}{c} \alpha_{x}∆T\\ \alpha_{y}∆T\\ 0 \end{array}\right\}zdz$$

Introducing a new matrix [D], Equation (15) can be rewritten as
(17)$$\left\{\begin{array}{c} M_{xx}\\ M_{yy}\\ M_{xy} \end{array}\right\}+\left\{\begin{array}{c} M_{xx}^{th}\\ M_{yy}^{th}\\ M_{xy}^{th} \end{array}\right\}=\left[\begin{array}{ccc} B_{11} & B_{12} & B_{16}\\ B_{12} & B_{22} & B_{26}\\ B_{16} & B_{26} & B_{66} \end{array}\right]\left\{\begin{array}{c} \varepsilon_{xx}^{0}\\ \varepsilon_{yy}^{0}\\ \gamma_{xy}^{0} \end{array}\right\}+\left[\begin{array}{ccc} D_{11} & D_{12} & D_{16}\\ D_{12} & D_{22} & D_{26}\\ D_{16} & D_{26} & D_{66} \end{array}\right]\left\{\begin{array}{c} k_{xx}\\ k_{yy}\\ k_{xy} \end{array}\right\}$$

The material coefficients (*Aij*, *Bij*, *Dij*) are known as the extensional stiffness, the extension-bending coupling stiffness, and the bending stiffness coefficients, respectively. These are given by
(18)$$\left(A_{ij},\;B_{ij},\;D_{ij}\right)=\int_{-h/2}^{h/2}{\bar{Q}}_{ij}\left(1,\;z,\;z^{2}\right)dz\;$$

The laminate constitutive relations can now be written in compact form as
(19)$$\left\{\begin{array}{c} \left\{N\right\}\\ \left\{M\right\} \end{array}\right\}=\left[\begin{array}{cc} \left[A\right] & \left[B\right]\\ \left[B\right] & \left[D\right] \end{array}\right]\left\{\begin{array}{c} \left\{\varepsilon^{0}\right\}\\ \left\{k\right\} \end{array}\right\}$$
where {*N*}, {*M*} include thermal resultants.

Using the laminate constitutive relations described earlier, laminate design of boat hull was performed based on laminate theory considering on thermal effect [10,11,12]. The structural safety was investigated through the final structural analysis to determine the design outcome.

The aramid fibers were applied as the material used in the structural design. The NCF (non-crimp fabric) weaving method was adopted for the weaving method. Figure 5 shows schematic diagram of NCF fabric structure. This method could set fibers at any angles without bending, and make fabric layers of various angles by the stitching method. In addition, mechanical property of material would be improved more than 20% compared to the woven fabric when manufacturing composites. This method is advantageous to applying the resin transfer molding method during the composite manufacturing process. The applied resin was polyester resin. The structural design result was determined as two kinds of laminated forms. Table 1 shows structural design result. Table 2 shows mechanical properties of applied aramid material.

The structural design result was applied to conduct the finite element modeling. The boundary condition was applied as a simply supported structure. Therefore, stress was not concentrated by restraint. For the structural modeling, the effect of weight load was considered by assuming that the external force by buoyancy and the weight load keep static balance. Through the structural analysis, stress was confirmed and the structural strength was evaluated.

As a structural analysis result of [0°/90°]_5_ which is the first design result, the maximum stress in the x-axis direction was confirmed as 246 MPa. The second design result [0°/45°/90°]_5_ was calculated as 147 MPa as a structural analysis result. The maximum stress in the y-axis direction was calculated 420 MPa and 231 MPa, respectively. The maximum tensile stress of NCF materials obtained from experiments was 473 MPa, which was found that both design results were valid. In this study, the second design result with less stress was determined as the final structural design result.

## 3. Resin Infusion Simulation

The resin infusion is an important variable for reasonable production when applying the RTM manufacturing method to manufacture. The trial-and-error method, which derives a valid one by manufacturing in various ways, is an infallible one. However, to reduce the cost of materials and experiments, the simulation method with numerical analysis is a significantly advantageous method. In this study, therefore, the resin infusion simulation software was used to perform numerical analysis. The resin infusion simulation was performed using Polyworx RTM-Worx simulation solver, Lekdijk 52, 2865 LD Ammerstol, Netherlands.

The method adopted to manufacturing structures in this study is the RTM which is a resin infusion method. Recently the RTM method has come into great prominence because of considering improvement of large composite structure molding and productivity [13]. The RTM method is a composite molding one that makes a preform, which is a fiber component, in advance and laminates on a mold to infuse resin. The optimized resin inlet and outlet positions were determined through the simulation that permeates resin into the earlier designed structural design molding model of the boat which is the target structure.

Considerations when infusing resin in the RTM method are the viscosity of resin, resin flow speed, resin infusion pressure and resin discharge vacuum pressure. In particular, important parts of the RTM method are proper resin infiltration pathways and resin infiltration time. In other words, the positions of inlet and outlet for resin infiltration are key variables affecting the manufacturing process. Therefore, it is important that the infusion path is accurately understood through the resin flow analysis. As most of the product shapes were complex, the resin flow analysis was performed to select the positions of inlet and outlet considered for mold design when manufacturing composite structures and to determine optimal molding conditions. In this study, it was applied a commercial finite element analysis program that could predict resin flow. To analyze the resin flow, Darcy’s law, the flow law of viscous fluids presented in equation 20, was applied to measure impregnation [14,15].
(20)$$t=\frac{\hat{A}\mu l^{2}}{2kP^{r}}$$
where, *t* is permeability time, $\hat{A}$ is dimension of fiber preform, *μ* is viscosity of resin, *l* is flow stream, *k* is permeability coefficient, $P^{r}$ is pressure difference.

Before applying the RTM method to simulate the hull, it was verified through plain plates. After performing the plate lamination simulation, it was compared through actual lamination. The specification of plate structure for simulation was shown in Table 3. This model was compared as a simple form that infused from the left and discharged to the right. As a result of analyzing the resin infiltration of plain plates, the total infiltration time was calculated as 414 s. The resin flow analysis result of plate was shown in Figure 6. The manufacturing process of plate by resin transfer was presented in Figure 7. Actual infiltration result of plain plates through manufacturing was found as 435 s. Therefore, error was analyzed as 5%.

The analysis and manufacturing result of simple plain plates were used to divide the resin infusion and discharge strategy for plain plates into three kinds to analyze them. There are three methods as Figure 8 for the resin infusion strategy of plain plates. For large plain plates, discharge channels are placed at the one edge and infusion channels are arranged at equidistant intervals as Figure 8a. For small-size plain plates, discharge channels are placed at every edge and infusion channels are arranged at the center as Figure 8b. In Figure 8c, they are placed oppositely.

The resin infusion method for plain plates was analyzed to study a method to apply to three-dimensional structures such as box-shaped or hemispheric-form hulls. For three-dimensional structures, such as box shapes or hulls, Figure 8b,c have some differences by the height variable. In this study, the resin infusion method for three-dimensional structures was analyzed to derive the optimized infusion method for hulls.

The boat hull was modeled to perform the resin flow analysis. Figure 9 shows modeling of hull structure for resin flow analysis. First, a single inlet was placed, and outlets and discharge pathways were applied as a shape surrounding the edge. Six outlets were placed. The diameter of inlet and outlet was set as 12 mm. As a result of the flow analysis, the total impregnation time was calculated 183,000 s, as shown in Figure 10. If the curing time of resin was considered, curing was progressed before impregnation was ended.

In the first case with a single inlet, because the infusion time is taken very long, it was improved with a method of increasing the number of inlets. In the second case, a total of seven resin infusion pathways were placed at intervals of 30cm, the discharge pathways were arranged as a shape surrounding the edge, and six outlets were placed. The result is as shown in Figure 11, and the total impregnation time was 60,900 s.

In the third case, a total of seven infusion pathways were arranged at intervals of 30 cm like the second case, two inlets were placed per infusion pathway. The discharge pathways were arranged as a shape surrounding the edge, and six outlets were placed. The diameter of inlet and outlet was set upward as 18 mm. The result is as Figure 12, and the total impregnation time was 3,640 s. Additionally, when three inlets were placed at an infusion pathways, its total impregnation time was not shortened compared to the case of two inlets. Additionally, when decreasing the interval between infusion pathways to 20 cm, the total impregnation time was reduced by about 5%, but in this case, because the cost of subsidiary materials rises, the efficiency was found not high compared to cost increase.

In this study, optimal inlet and outlet were determined through the resin flow analysis of the final boat hull structure. A total of seven inlets were applied at the lower part of the hull. For outlets, it was confirmed optimal that six outlets were applied as a shape surrounding the edge. It was finally determined that infusion pathways were placed at intervals of 30 cm on the inside and two outlets were applied at every discharge pathway. The actual impregnation time taken for lamination work was a total of 3778 s, and it was analyzed that the error was 3.6% compared to the numerical analysis result. It was found that there was no non-impregnated part inside the structure and it was uniformly impregnated.

## 4. Manufacturing of Prototype

Above numerical analysis result was used to consider optimal resin infusion time and select the positions of inlet and outlet. The resin flow analysis result of the RTM manufacturing process was reflected to determine the positions of the resin inlets and outlets to be at the bottom and edge of the hull, respectively. Considering the positions of resin inlet and outlet, the design and manufacturing of molds was conducted to make the prototype. The boat structure design shape was reflected to produce a master model and then to make the mold. For materials used in producing the master mold, usually used are metallic materials, resin, and composite, etc. Since the master mold made with composites does not cause a decline in quality, even after de-molding many times, it is often used in the area which produces high-quality products. Accordingly, CNC 5-axis machining was used to manufacture the master mold. Figure 13 is a photo of using 5-axis machining to make the wooden pattern, and Figure 14 shows a shape for making the final mold to apply gelcoat.

After making the mold, it was used to produce the prototype. Based on the design result, aramid fibers were laminated and the RTM equipment was mounted. The RTM method was used to make respective structure components and then combine finally to complete the prototype. Figure 15, Figure 16 and Figure 17 show detailed manufacturing processes.

## 5. Test and Evaluation

In this study, the weight of the prototype manufacturing was measured and evaluated to compare with the existing glass fiber product. In initial hull weight lightening effect analysis, the target weight of hull was set as 495 kg. The error was set as about 10%. As a result of manufacturing, actual weight was 510 kg, which was different by 3%, so the target weight was achieved. The weight of the existing boat applying GFRP and the boat applying aramid fibers was compared. The compared result is presented in Table 4. The case of applying aramid fibers achieved weight lightening of 36% compared to GFRP. This leads to being able to increase in fuel efficiency and lengthen boat’s service life. The performance was evaluated by making a trial run of the final prototype. It was evaluated as an excellent family boat in the aspect of the hull’s condition, sliding ability, and structure. Figure 18 is a scene where evaluates 15° turning ability of the prototype.

## 6. Conclusions

In this study, aramid composite was applied to conduct structural design and analysis of small boats. The structural design load of boat was analyzed to perform structural design. Structural design result was analyzed to determine the final thickness through structural safety examination.

The structural design result of boats was reflected to manufacture the final prototype. The RTM manufacturing method was adopted for prototype manufacture. The RTM process is a method that infuses resin, in which the time of permeating into fibers by resin flow is acting as an important variable in the molding time. In this work, the inlet and outlet were selected and the optimal molding conditions were derived, to be considered for the composite structure manufacturing process through the resin flow analysis of the RTM method. Comparing the resin flow analysis result with the experimental result, success or failure of the molding could be measured within the limiting condition, and the optimized manufacturing method was established. In addition, the weight was reduced through weight lightening compared to the glass fiber structure.

## Figures and Tables

**Figure 1 materials-14-05918-f001:**
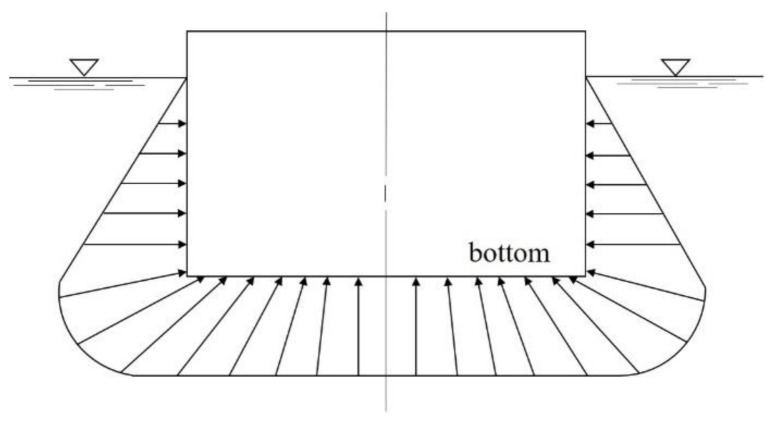
The configuration of hull structure.

**Figure 2 materials-14-05918-f002:**
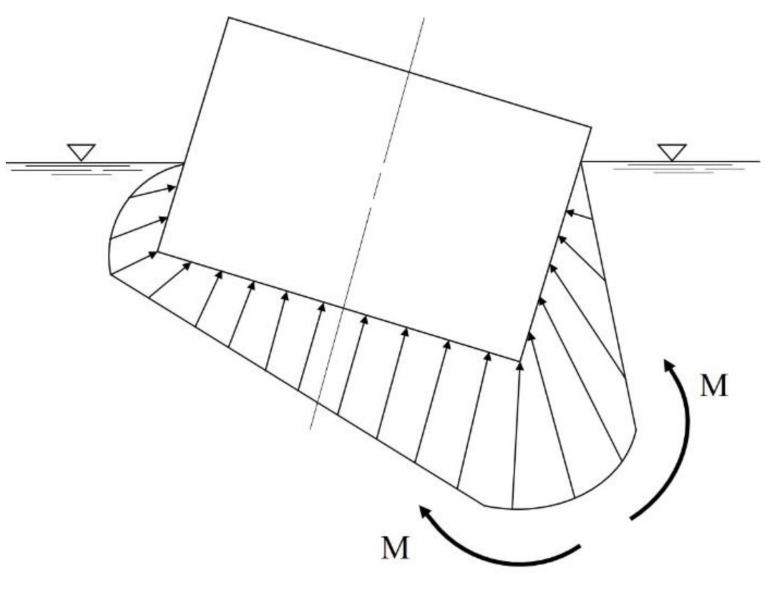
Load diagram of hull applied to moment.

**Figure 3 materials-14-05918-f003:**
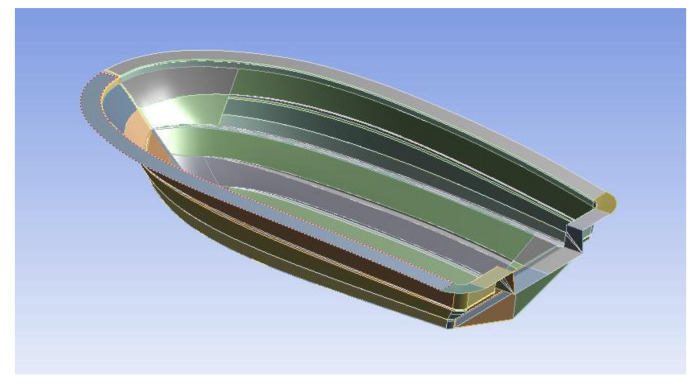
The configuration of hull structure.

**Figure 4 materials-14-05918-f004:**
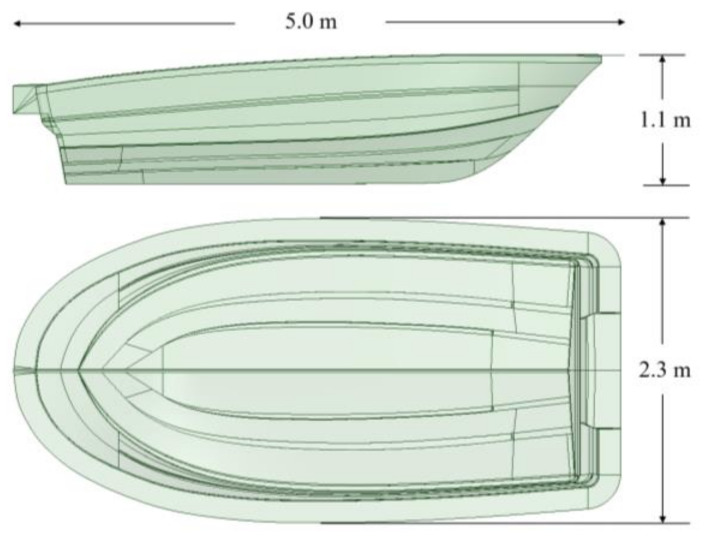
The dimension of boat hull.

**Figure 5 materials-14-05918-f005:**
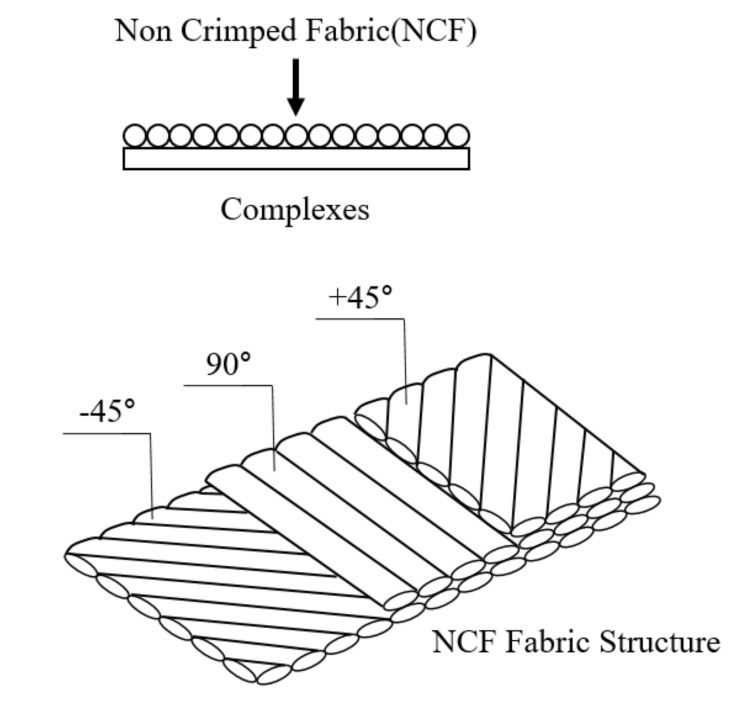
The dimension of boat hull.

**Figure 6 materials-14-05918-f006:**
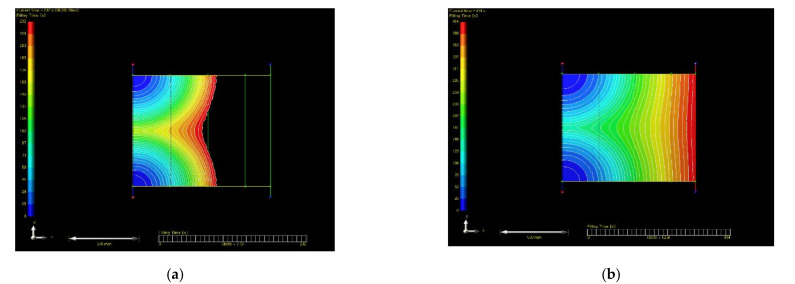
The resin flow analysis result of plate.: (**a**) t = 232 s (56% filled); (**b**) t = 414 s (100% filled).

**Figure 7 materials-14-05918-f007:**
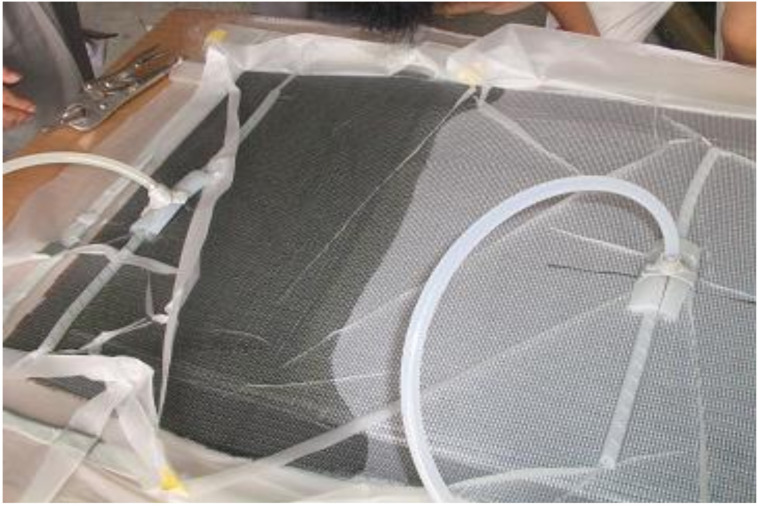
Manufacturing process of plate by resin transfer.

**Figure 8 materials-14-05918-f008:**
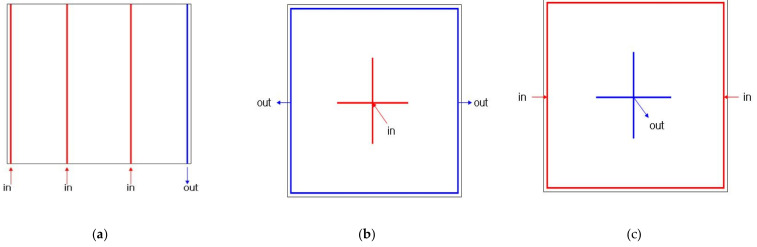
The resin infusion strategy of plates: (**a**) Injection from three parts and exhaust from one part, (**b**) Inject from the center and exhaust from the side, (**c**) Injection from the side and exhaust from the center.

**Figure 9 materials-14-05918-f009:**
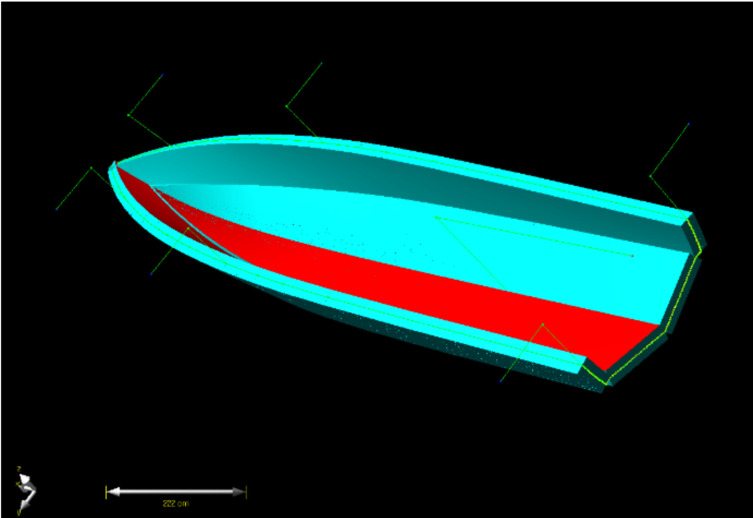
Modeling of hull structure for resin flow analysis.

**Figure 10 materials-14-05918-f010:**
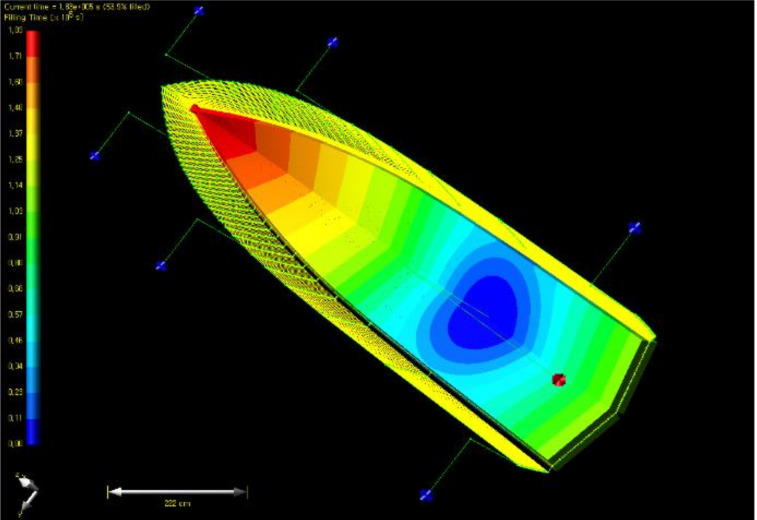
Resin flow analysis result of case 1 (filling time).

**Figure 11 materials-14-05918-f011:**
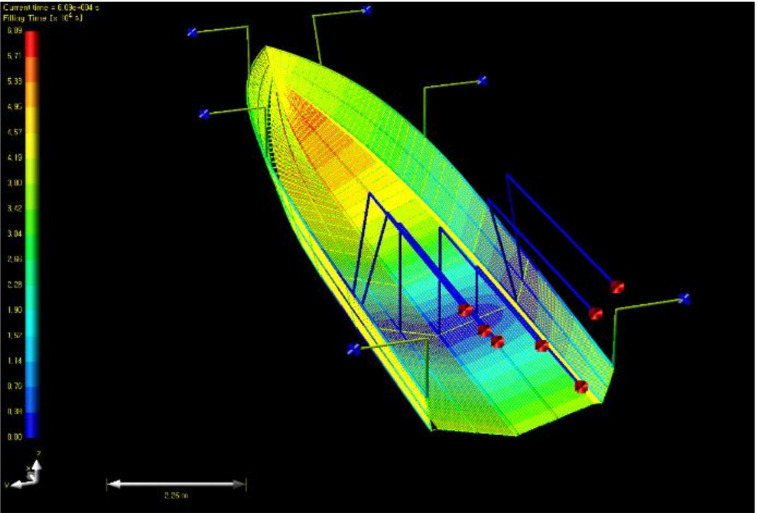
Resin flow analysis result of case 2 (filling time).

**Figure 12 materials-14-05918-f012:**
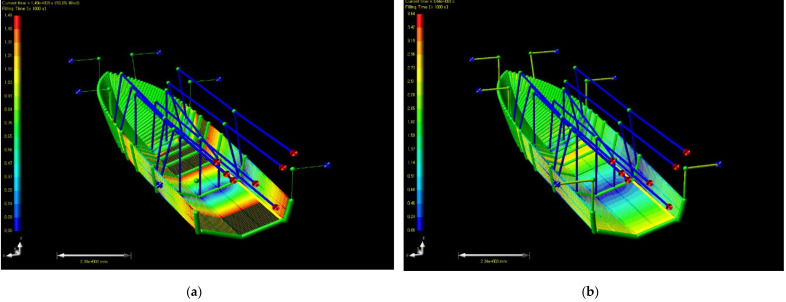
Resin flow analysis result of case 3 (filling time): (**a**) 50% filled; (**b**) 100% filled.

**Figure 13 materials-14-05918-f013:**
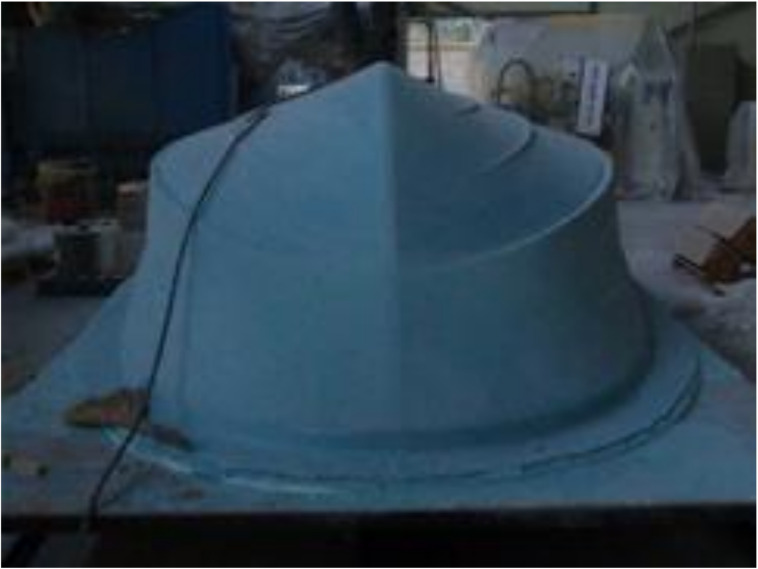
Manufacturing of mock up by five-axis machining.

**Figure 14 materials-14-05918-f014:**
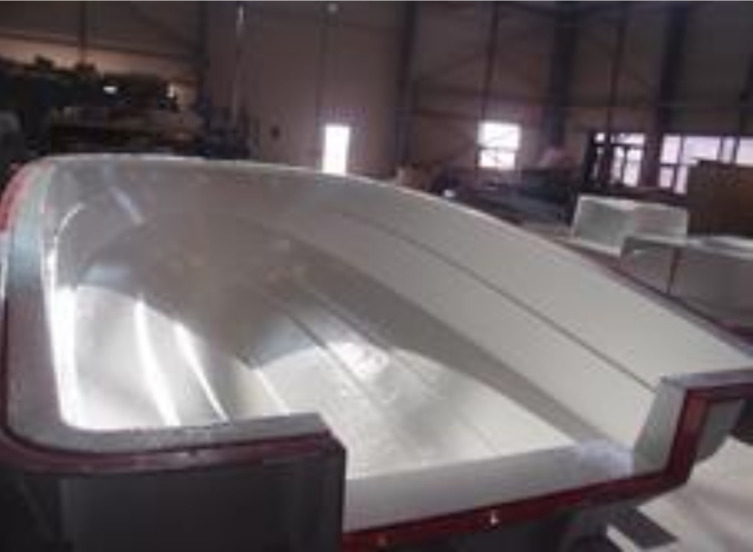
Application of gelcoat.

**Figure 15 materials-14-05918-f015:**
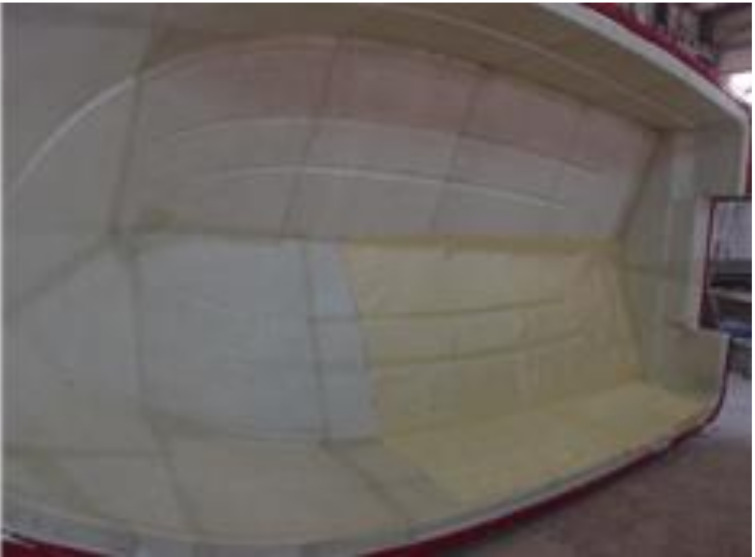
Laminate of hull structure.

**Figure 16 materials-14-05918-f016:**
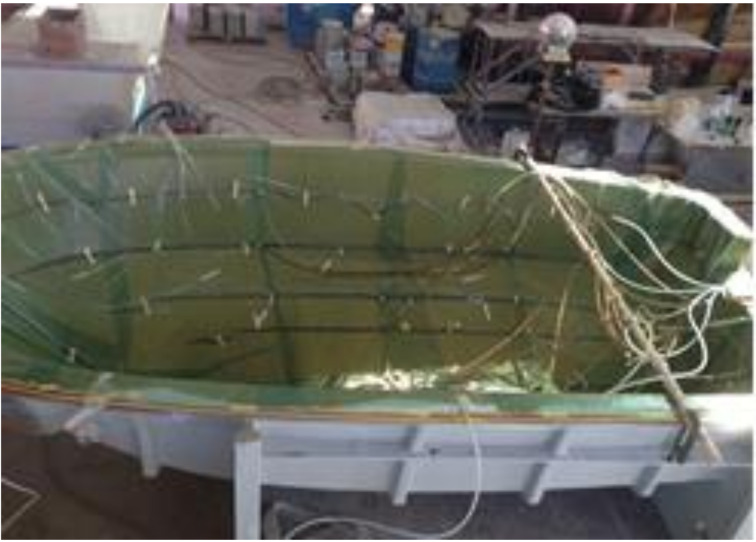
RTM process of laminate structure.

**Figure 17 materials-14-05918-f017:**
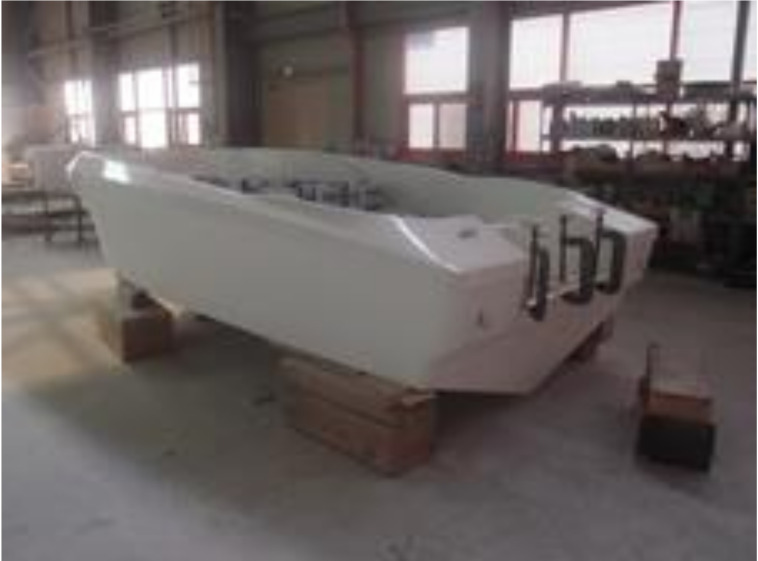
Final manufactured prototype.

**Figure 18 materials-14-05918-f018:**
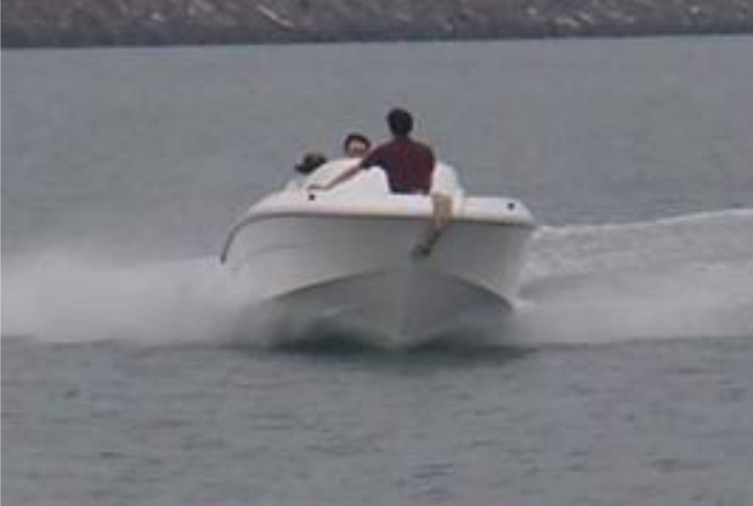
Test and evaluation of manufactured boat.

**Table 1 materials-14-05918-t001:** Structural design result of hull.

Case No.	Material Type	Orientation	Total Thickness (mm)
1	NCF	[0°/90°] _5_	2.5
2	NCF	[0°/45°/90°] _5_	3.75

**Table 2 materials-14-05918-t002:** Mechanical properties of aramid material.

E_1_ (MPa)	E_2_ (MPa)	ν_12_	G_12_ (MPa)	G_13_ (MPa)	G_23_ (MPa)
75,000	6000	0.34	2000	2000	2000

**Table 3 materials-14-05918-t003:** Specification of plate structure for simulation.

Item	Specification
Stacking sequence	[0°]_2_
Dimension	1100 mm × 1040 mm

**Table 4 materials-14-05918-t004:** Structural design result of hull.

**Structural Part**	**Aramid Fiber Boat (kg)**	**GFRP Boat (kg)**	**Lightweight (%)**
Hull	150	202	34
Construction	110	150	36
Deck	115	150	30
House	50	62	24
Tank and hardware	70	110	37
Total	495	674	36

## Data Availability

The data presented in this study are available on reasonable request from the corresponding author.

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
