# Peer review of "Study on Structural Design and Analysis of Composite Boat Hull Manufactured by Resin Infusion Simulation"

_materials, 2021, doi:10.3390/ma14205918_

Round 1

Reviewer 1 Report

The article has a lot of mistakes.

The Introduction must be rewritten as it presents repetitions of sentences, saying in a very simple way some reference articles. It did not present the reason for studying RTM manufacturing.

There are grammatical errors, for example on line 66.

On page 4, who developed this theory? Was this theory elaborated in this work?

What software was used for the simulation?

In line 134, the use of aramid fiber was presented. This is one of the big mistakes in the article. Aramid fiber was not presented in the abstract, introduction, materials and methods and was highlighted in the conclusions. This makes the article must be rewritten in a cohesive way.

Author Response

The article has a lot of mistakes.

The Introduction must be rewritten as it presents repetitions of sentences, saying in a very simple way some reference articles. It did not present the reason for studying RTM manufacturing.

→ We revised the manuscript following reviewer’s comment. The introduction has been restructured and rewritten.

In this paper, structural design and analysis of composite boat hull was performed. Resin trans-fer molding manufacturing method was adopted for manufacturing the composite boat hull. The RTM process is an advanced composite manufacturing method that allows much higher quality product than the hand lay-up process, and less manufacturing cost compared to the autoclave method. Therefore, the RTM manufacturing method was adopted. The mechanical properties of the various aramid fibers and polyester resin were investigated. Based on this, structural design of boat hull was performed using glass fiber/polyester. After structural design, the optimized resin infusion analysis for RTM manufacturing method was performed. Through the resin infusion analysis, it is confirmed that the designed location of resin injection and outlet is acceptable for manufacturing.

There are grammatical errors, for example on line 66.

→ We revised the whole manuscript carefully to avoid language errors. In addition, we asked several colleagues who are skilled native to check the English. We believe that the language is now acceptable for the review process.

The buoyancy force due to water pressure was applied as an external load. The buoyant force is the greatest at the bottom of a hull.

On page 4, who developed this theory? Was this theory elaborated in this work?

→ This theory is applied to the design of composite materials. This is the classical lamination theory defined in the previous study. In this work, this theory was applied to design of composite laminated pattern.

What software was used for the simulation?

→ We revised the manuscript following reviewer’s comment.

The resin infusion simulation was performed using Polyworx RTM-Worx FEM flow simulation solver.

In line 134, the use of aramid fiber was presented. This is one of the big mistakes in the article. Aramid fiber was not presented in the abstract, introduction, materials and methods and was highlighted in the conclusions. This makes the article must be rewritten in a cohesive way.

→ We revised the manuscript following reviewer’s comment. It has been modified to apply aramid fibers to the manuscript.

The composite materials are widely used in vehicle design because of their excellent strength and light weight. Although the aramid fiber has less strength than the high strength fiber such as the carbon fiber, it has low manufacturing cost. Accordingly, it can be applied as very advantageous composite when an appropriate resin has been selected. In this work, structural design and analysis of composite boat hull was performed using aramid fiber/polyester.

Reviewer 2 Report

This paper can be considered afer minor revision.

In the abstract, the authors only refered to the glass fiber, but in test and evaluation section,  both the aramid fiber and glass fiber  are refered.

Author Response

This paper can be considered afer minor revision.

In the abstract, the authors only refered to the glass fiber, but in test and evaluation section, both the aramid fiber and glass fiber are refered.

→ We revised the manuscript following reviewer’s comment. It has been modified to apply aramid fibers to the manuscript.

In this paper, structural design and analysis of composite boat hull was performed. Resin trans-fer molding manufacturing method was adopted for manufacturing the composite boat hull. The RTM process is an advanced composite manufacturing method that allows much higher quality product than the hand lay-up process, and less manufacturing cost compared to the autoclave method. Therefore, the RTM manufacturing method was adopted. The mechanical properties of the various aramid fibers and polyester resin were investigated. Based on this, structural design of boat hull was performed using glass fiber/polyester. After structural design, the optimized resin infusion analysis for RTM manufacturing method was performed. Through the resin infusion analysis, it is confirmed that the designed location of resin injection and outlet is acceptable for manufacturing.

Reviewer 3 Report

This paper  performs structural design and analysis of composite boat 
hull. Composite layup theories are used to design the structures. The RTM processing is introduced for manufacturing.  The manufacturing process  for resin flow behaviors are simulated by FEA. This paper is a little mixed. I want to say the design for composite hull should consider more factors such as hydrodynamic behaviors and even impact and fatigue resistance. In addition, strength should evaluated after layup design. These factors are not neglected.  In summary, this paper lack the introduction ande analysis of a deep and comprehensive theoretical method for design work. 

Author Response

This paper performs structural design and analysis of composite boat hull. Composite layup theories are used to design the structures. The RTM processing is introduced for manufacturing. The manufacturing process for resin flow behaviors are simulated by FEA. This paper is a little mixed. I want to say the design for composite hull should consider more factors such as hydrodynamic behaviors and even impact and fatigue resistance. In addition, strength should evaluated after layup design. These factors are not neglected. In summary, this paper lack the introduction and analysis of a deep and comprehensive theoretical method for design work.

→ We revised the manuscript following reviewer’s comment. The manuscript was revised by adding scientific theory and consideration. The manuscript was improved by adding detailed novelty and contribution in the text and figure.

In this study, boat design was performed by analyzing buoyancy and various forces. However, the impact and fatigue was not taken into account. We will add it to future research. This study mainly focused on applying aramid fibers and optimizing them through resin injection simulation.

Round 2

Reviewer 1 Report

The authors made the necessary modifications for publication.

Reviewer 3 Report

I recommend the publication of this paper in the present version